# Geodesics to characterize the phylogenetic landscape

**Marzieh Khodaei**[1]*, **Megan Owen**[2], **Peter Beerli**[1]

**1** Department of Scientific Computing, Florida State University, Tallahassee, FL, United States of America,
**2** Department of Mathematics, Lehman College and Graduate Center, CUNY, NY, NY, United States of America

* mk16e@fsu.edu

**Data Availability Statement:** We implemented our method as free software named PATHTREES under the MIT open-source license. The source code and the documentation of PATHTREES are available at https://github.com/TaraKhodaei/PATHTREES.git The primate data is available from the Tutorial

## Abstract

Phylogenetic trees are fundamental for understanding evolutionary history. However, finding maximum likelihood trees is challenging due to the complexity of the likelihood landscape and the size of tree space. Based on the Billera-Holmes-Vogtmann (BHV) distance between trees, we describe a method to generate intermediate trees on the shortest path between two trees, called pathtrees. These pathtrees give a structured way to generate and visualize part of treespace. They allow investigating intermediate regions between trees of interest, exploring locally optimal trees in topological clusters of treespace, and potentially finding trees of high likelihood unexplored by tree search algorithms. We compared our approach against other tree search tools (PAUP*, RAxML, and REvBAYES) using the highest likelihood trees and number of new topologies found, and validated the accuracy of the generated treespace. We assess our method using two datasets. The first consists of 23 primate species (CytB, 1141 bp), leading to well-resolved relationships. The second is a dataset of 182 milksnakes (CytB, 1117 bp), containing many similar sequences and complex relationships among individuals. Our method visualizes the treespace using log likelihood as a fitness function. It finds similarly optimal trees as heuristic methods and presents the likelihood landscape at different scales. It found relevant trees that were not found with MCMC methods. The validation measures indicated that our method performed well mapping treespace into lower dimensions. Our method complements heuristic search analyses, and the visualization allows the inspection of likelihood terraces and exploration of treespace areas not visited by heuristic searches.

## 1 Introduction

Evolutionary trees, or phylogenetic trees, have been used extensively throughout systematic biology and other fields to represent the evolutionary history of species. How to compute the best tree and how to characterize the uncertainty of estimates of the branch lengths and the topology is an on-going challenge. Generally tree search methods seek a globally best tree under some optimization criteria (e.g. parsimony [1], distance methods [2, 3], or maximum likelihood [4]), but the number of potential trees grows exponentially relative to the number of

website of RevBayes: https://revbayes.github.io/tutorials/ctmc/#example-character-evolution-under-the-jukes-cantor-substitution-model (the dataset url is https://revbayes.github.io/tutorials/ctmc/data/primates_and_galeopterus_cytb.nex). The snake dataset is available from the Dryad Digital Repository: http://dx.doi.org/10.5061/dryad.7hs34mj.

**Funding:** PB: National Science Foundation, DBI 1564822 and DBI 2019989 MO: National Science Foundation, DMS 1847271 The funders had no role in study design, data collection and analysis, decision to publish, or preparation of the manuscript.

**Competing interests:** The authors have declared that no competing interests exist.

leaves [5]. Furthermore, it is NP-hard to compute the maximum likelihood tree [6, 7] or most parsimonious tree [8], so except for small numbers of taxa where exhaustive search is possible, heuristic methods must often be used to search the treespace. These methods explore locally best trees in the hope that the best local tree found is equivalent to the global best tree (e.g. maximum likelihood programs RAxML [9], PHYML [10], and PAUP* [11]). Similarly, Bayesian inference programs use tree rearrangement moves to generate proposals to estimate the posterior probability (e.g. Bayesian inference programs MRBAYES [12], REVBAYES [13], and BEAST [14]).

Initially, many optimality criteria to compare trees were developed, the most prominent being parsimony [1], distance methods [2, 3], and maximum likelihood [4]. Computational power has increased considerably since then, so computationally intense probabilistic methods such as maximum likelihood and Bayesian inference have supplanted faster but less accurate methods. Researchers now commonly use maximum likelihood programs, such as RAxML [9], PHYML [10], and PAUP* [11], or Bayesian inference programs, such as MRBAYES [12], REVBAYES [13], and BEAST [14].

Both Bayesian inference and maximum likelihood methods use random changes in the tree topology to search the treespace. Even when we record all visited trees in a maximum likelihood search or look at all collected trees in a Bayesian Markov chain Monte Carlo run, the trees are not evenly distributed throughout treespace: some regions are heavily sampled, while other regions are sampled sparsely or not at all (for example, see Fig 2). While treespace is much too large to sample entirely, an MCMC may not explore the space or posterior distribution efficiently due to revisiting trees topologies [15] or due to low posterior nodes separating peaks [16]. While in an ML tree search, Money and Whelan [17] show that different rearrangement moves correspond to discretized treespaces with different numbers of local optima.

While the concept of treespace is often used informally to mean the set of all possible phylogenetic trees meeting some condition, such as having $n$ leaves, a treespace can be formally defined as a discrete or continuous metric space (ie. a geometric space with a distance measure between points meeting certain conditions) where each point corresponds to a tree (see [18] for a comprehensive review of treespaces). In this paper, we use the Billera-Holmes-Vogtmann (BHV) treespace [19], which is a continuous, piece-wise Euclidean space containing all trees with branch lengths and $n$ leaves. This space contains unique shortest paths, or geodesics, between any two points, with the lengths of these paths being the Billera-Holmes-Vogtmann (BHV) distance. Both the BHV distance and geodesics between trees can be computed in polynomial time [20]. The weighted Robinson-Foulds distance (wRF) [21] corresponds to using an $L_1$ metric instead of an $L_2$ metric on the piecewise-Euclidean orthants of BHV treespace. The wRF distance does not have unique geodesics but is faster to compute than the BHV distance and is at most a multiplicative factor of $\sqrt{2}$ larger than the BHV distance [22]. The Robinson-Foulds (RF) distance [23] is the same as the weighted Robinson-Foulds distance when all edge lengths are set to be 1.

A landscape is a configuration space or a metric space of trees (with or without branch lengths) $\mathcal{T}$ and an associated real-valued cost or fitness function $f : \mathcal{T} \rightarrow \mathbb{R}$. Landscapes were first defined by Bastert et al. [24] on trees without branch lengths, so the metric space $\mathcal{T}$ was a graph. A landscape on trees without branch lengths can be visualized as a colored graph, where the trees are the nodes, colored by the fitness function value, and edges represent a minimal rearrangement move between trees, such as Subtree-Prune and Regraft (SPR) [16]. Alternatively, for trees with or without branch lengths, distances can be computed between the trees, and visualized in 2 or 3 dimensions using Multi-Dimensional Scaling (MDS). MDS approximates the pairwise distances between points by mapping them in a lower-dimensional

Euclidean space [25], and was first applied to sets of phylogenetic trees by Amenta and Klingner [26] and popularized by Hillis et al. [27]. Trees in an MDS visualization can be colored by a qualitative descriptor (e.g. associated cluster [28–30], topology [28] or gene [27]) or by a quantitative descriptor (e.g. likelihood [27], posterior probabilitiy [31], or minimum implied gap (MIG) score, which measures congruence with the fossil record [32]). Different tree distance metrics have also been used, including RF [27, 29, 32], wRF [27], the Kendall-Colijn metric [28, 30], BHV [29], and Nearest-Neighbor Interchange [31].

We use MDS to visualize landscapes where the fitness function is the log likelihood, but approach the visualization in a different way from previous work. For a given area of interest in treespace, such as the region bounded by the trees from an MCMC run, we sample trees along certain geodesics crossing this area to get a representative set of trees ("pathtrees"). We then use MDS to map these pathtrees into 2 dimensions, color them by their likelihood or topology, with the shade determined by their optimized log likelihood, and use interpolation to estimate the likelihood landscape in the treespace area of interest. We also show the landscape function as a 3-D surface over the 2-D MDS plot and call this a likelihood surface. Thus, our visualizations try to illustrate the overall landscape tendency in the part of treespace of interest, instead of only point values of the fitness function, possibly unevenly distributed.

There are several recent packages or programs for visualizing trees using MDS, but they focus on coloring the points by cluster rather than a fitness function. TREESCAPER [33] is a standalone GUI that allows different tree distance functions, cost functions for dimensionality reduction, and non-linear dimension reduction algorithms to be used. TREESPACE [30] is an R package that allows a wide variety of tree metrics and methods for clustering trees to be used. Smith [34] evaluated the performance of multiple aspects of low-dimensional representations of sets of trees, and provides an R package TreeDist for users to do the same. Finally, R We There Yet (RWTY) [35], a package for analyzing Bayesian analyses convergence, can produce nonlinear MDS visualization of landscapes using the RF or path difference distance [36] and colored by the likelihood. Some authors [34, 37, 38]) have analyzed how well MDS visualizes treespace, and suggested validation measures. There are other ways to visualize sets of related trees beyond dimensionality reduction, such as super-imposing the trees on each other, as in DensiTree [39], or sophisticated tree comparison visualizers, like ADView [40].

The audience for our method and software are biologists who need to analyze their data and generate trees and describe potential alternative topologies and need to be able to compare the best tree found by heuristic searches or Bayesian inference with potential alternatives. We will focus on likelihood as the optimality criterion for the rest of this paper. While there is a closed-form expression to compute the likelihood of a given tree, given sequence data, the likelihood function itself is very complex with multiple local optima [41, 42]. Finding the best tree in the presence of multiple local and global optima and the presence of regions of trees with similar, high likelihood, such as islands [43, 44] and terraces [45, 46] is difficult. Our method can deliver additional support for other heuristic methods by investigating the relationship among trees in the BHV space and visualizing the landscape at different scales in an area of interest in treespace using MDS. We apply our method to two datasets, and discover novel high likelihood tree topologies. We distribute our approach in the Python package PATHTREES.

## 2 Materials and methods

We developed a method to generate and visualize the log-likelihood landscape in an area of interest in a treespace, which can sometimes find trees of high likelihood unexplored by tree search algorithms. We sample trees ("pathtrees") along the shortest paths in BHV treespace between points on the vertices of a convex hull enclosing the area of interest, and also compute

the optimized branch lengths for each tree topology found. We then use MDS to map all of these trees into two dimensions and use splines to interpolate the log-likelihood in the spaces between trees to visualize the relationship of the trees and their landscape. For the rest of this paper, we use "treespace" to refer to the Billera-Holmes-Vogtmann (BHV) treespace, unless otherwise specified.

## 2.1 Billera-Holmes-Vogtmann (BHV) treespace and shortest paths

The Billera-Holmes-Vogtmann (BHV) treespace models all phylogenetic trees with a fixed set of leaves. It is formed from a set of Euclidean regions, called orthants. Each orthant contains only trees with the same topology. Each tree topology consists of a unique set of splits [47], and each of these splits is assigned to one of the dimensions of the orthant. Each branch length in a tree becomes the coordinate in the orthant along the dimension corresponding to the branch's split. Two orthants with corresponding tree topologies that share splits are adjacent in the treespace, and their shared boundary orthant contains all trees containing exactly the shared splits. All orthants contain the origin, which corresponds to the star tree, so the space is connected.

The length of a path between two trees is computed by measuring the Euclidean length of the path in each orthant it passes through, and summing those lengths. There is a unique shortest path, or geodesic, connecting two phylogenetic trees $T_1$ and $T_2$ in BHV treespace [19], and it can be computed in polynomial time $O(n^4)$, where $n$ is the number of leaves in the trees, by the Geodesic Treepath Problem (GTP) algorithm [20].

If trees $T_1$ and $T_2$ have no common splits, then the GTP algorithm starts with a simple initial path, called the cone path, which connects trees $T_1$ and $T_2$ to the origin (a star tree), and hence each other, by straight lines. Then, the cone path is transformed into a series of successively shorter paths until the geodesic is obtained. At each step, the algorithm identifies one new orthant that the current path can be modified to pass through to yield a shorter path. When trees $T_1$ and $T_2$ have splits in common, the algorithm first subdivides $T_1$ and $T_2$ along the common splits, and runs the GTP algorithm described above on each pair of subtrees. The shortest paths between the subtrees are then combined into the overall geodesic between $T_1$ and $T_2$. Fig 1 demonstrates an example of a geodesic between two trees, and the geometric representation of treespace. Fig 1A shows a starting tree ($T_1$), an ending tree ($T_2$), and the two trees where the geodesic between $T_1$ and $T_2$ crosses orthant boundaries. Fig 1B shows the geodesic, the cone path, parts of the three orthants that the geodesic passes through, and an example tree of each orthant. Moving along the geodesic from start tree $T_1$ to end tree $T_2$, the intermediate tree branches shrink to zero length at orthant boundaries, and new branches begin to grow.

## 2.2 Sampling trees on the shortest path between tree pairs

Our method samples trees along the shortest paths (geodesics) between points on the boundary of our area of interest. The topologies and edge lengths for these sampled trees on the geodesic between trees $T_1$ and $T_2$ are given by Theorem 2.4 of Owen and Provan [20] and Theorem 1.2 of Miller et al. [48]. First let $(A_1, A_2, \ldots, A_k)$ be a partition of the splits in $T_1$ that are not in $T_2$, where $A_i$ is the set of splits whose branches shrink to zero length at the $i$-th orthant boundary along the geodesic. Let $(B_1, B_2, \ldots, B_k)$ be a partition of the splits of $T_2$ that are not in $T_1$, where $B_i$ is the set of splits whose branches begin growing from zero length at the $i$-th orthant boundary. Let $C$ be the set of splits common to $T_1$ and $T_2$, and parameterize the geodesic between $T_1$ and $T_2$ by $0 \leq \lambda \leq 1$. For split $e$ in tree $T$, denote its branch length in $T$ by $|e|_T$ and for the set of splits $S$ in tree $T$, let $||S|| = \sqrt{\sum_{e \in S} |e|_T^2}$. Then by Theorem 2.4 of [20] and

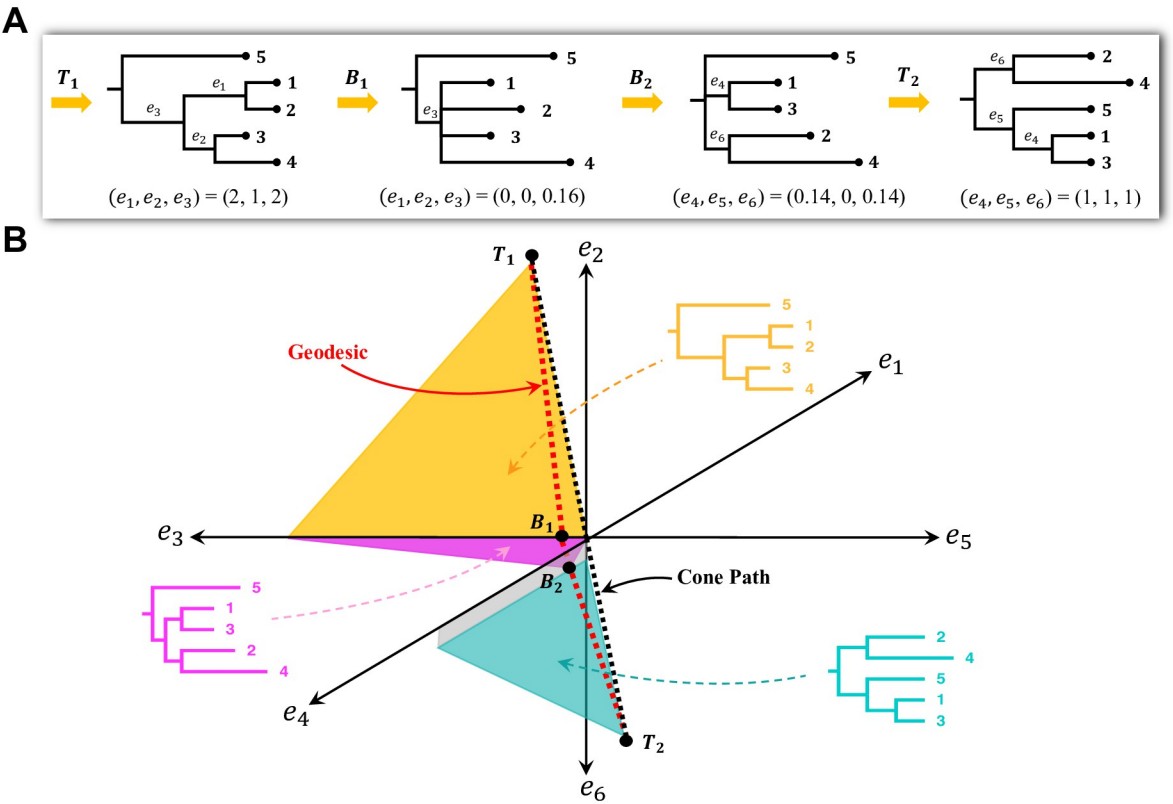

**Fig 1. Shortest path (geodesic) between two trees in the BHV treespace.** (A) A start tree $T_1$ and an end tree $T_2$ with two intermediate trees $B_1$ and $B_2$ on the boundaries of orthants, axis $e_3$ and the quarter-plane formed by axes $e_4$ and $e_6$, respectively. (B) Black dots mark the trees in Panel A; the cone path is the black dotted line; the geodesic is the red dotted line; each colored region is part of a different orthant containing the geodesic; and an example tree with arbitrary branch lengths is shown for each orthant.

Theorem 1.2 of [48], a tree $T_i$ on this geodesic at position $\lambda$ in the $i$-th orthant contains exactly the splits

$$\mathcal{S} = C \cup B_1 \cup \ldots \cup B_i \cup A_{i+1} \cup \ldots \cup A_k, \tag{1}$$

such that split $e \in \mathcal{S}$ has branch length

$$|e|_{T_i} = \begin{cases} \dfrac{(1-\lambda)||A_j|| - \lambda||B_j||}{||A_j||}|e|_{T_1} & e \in A_j \\[2ex] \dfrac{\lambda||B_j|| - (1-\lambda)||A_j||}{||B_j||}|e|_{T_2} & e \in B_j \\[2ex] (1-\lambda)|e|_{T_1} + \lambda|e|_{T_2} & e \in C \end{cases} \tag{2}$$

Thus, the trees along the geodesic between $T_1$ and $T_2$ can only contain splits from $T_1$ and $T_2$, and any split common to $T_1$ and $T_2$ appears in all trees along the geodesic between them. Additionally, as shown in Fig 1 where the geodesic passes through the quarter plane formed by the axes $e_3$ and $e_4$, a geodesic can non-trivially pass through lower-dimensional orthants, corresponding to trees with 0 length edges. Each tree $T_i$ generated along the path is then stored in the standard Newick format. Some trees on the geodesic are not bifurcating trees.

### 2.3 Finding the starting trees for PATHTREES

Our method visualizes the landscape in an area of interest in treespace. The boundary of this area is defined by a set of trees. We compute the shortest path between each pair in this set of trees, and sample trees along these paths to get the pathtrees. Our package can generate random starting trees, but with a larger number of taxa, these random trees span a very large section of treespace. Instead of random starting trees, we use a large set of trees generated by another method, for example, by REVBAYES using Markov chain Monte Carlo. We calculate the BHV distance between these trees and map them to two dimensions using MDS. The trees that are on the vertices of the convex hull, the smallest convex polygon enclosing all of the trees in the 2-D MDS plane, are then extracted and used as the starting trees for PATHTREES. The pathtrees will only depend on the trees on the convex hull. Ideally, we would want to calculate the hull in treespace and not the 2-D MDS space but currently, there is not an algorithm to achieve that. For example, we collected 50,000 trees using the program REVBAYES and the primate dataset (outlined in more detail in Section 2.7) and then extracted 1000 trees from the last 1/10 of the MCMC chain. Fig 2 shows the space of 1000 trees and the trees on the vertices of the corresponding convex hull.

### 2.4 Visualizing treespace and generating pathtrees

The largest orthants in the BHV treespace for $n$-leaf rooted trees have dimension $2n - 3$, so landscapes on this treespace cannot be visualized directly. Instead, we follow precedent [26] and we generate a distance matrix for all $N$ trees we wish to visualize, and use this distance matrix as input into a multidimensional scaling (MDS) algorithm [49] to compress the higher dimensional treespace into two dimensions. The BHV tree space is high dimensional and compressing this space using MDS to 2 dimension may bring unrelated trees close to each other. We evaluated this mapping by comparing the 2-D MDS coordinate distance matrix with the tree distance matrix using correlation measures, such as Pearson's $r$. We show the goodness of fit of a Shepard diagram [50] by calculating the correlation measures Pearson r, Spearman rho, and Kendall tau between the original distances and the MDS distances [34]. Since we are interested not only in the relationship among the trees but also in how well they fit the data, we calculate the log-likelihood for each tree and add this dimension to the 2-D MDS visualization either as contours or a third coordinate axis. For a smooth representation of the likelihood surface, we interpolate the log-likelihood values between the $N$ trees. We used two different methods: the cubic spline interpolation method [51] and the radial basis function (RBF) thin-plate spline interpolation [52]. The differences between the two interpolation methods (cubic spline interpolation and the thin-plate spline interpolation) are discussed in Section 3 in S1 File.

The MDS procedure is time-consuming for large distance-matrices. We experimented with two different distance metrics for visualization: the BHV distance and the weighted Robinson-Foulds (wRF) distance [21]. The wRF distance is faster to compute than the BHV distance and is, at most, a multiplicative factor of $\sqrt{2}$ larger than the BHV distance. In general, the tree distance used can have an effect on the MDS visualization, which is not well understood and an area of active research [27, 34].

The distribution of the sampled trees shown in Fig 2 highlights that some areas of treespace were sampled less well than others. In contrast, choosing trees along geodesics allow us to visualize trees that are evenly spaced between two arbitrary end-point trees. We demonstrate these pathtrees in Fig 3 where we selected three trees (colored triangles) from the 1000 sample trees visualized in Fig 2 and generated 20 trees on the shortest path in the BHV treespace between each pair of them and then visualized the contour and the surface of all 1060 trees in 2-D MDS space. The pathtrees bridge the gaps between the sampled trees. The selected three trees are

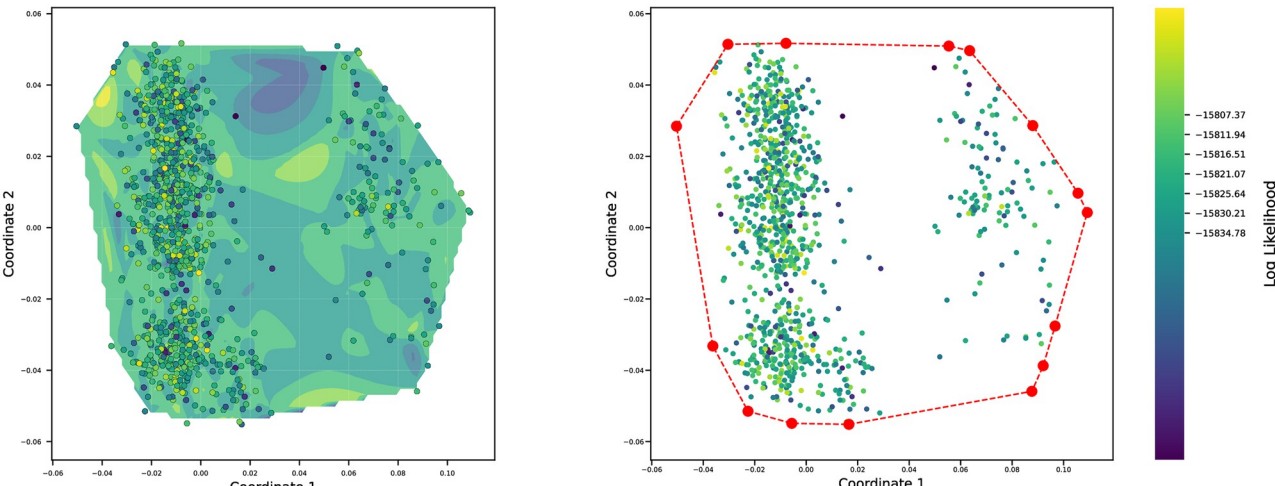

**Fig 2. An example of sample treespace and its convex hull.** Left: A log-likelihood contour plot of the first two multidimensional scaling (MDS) coordinates of the sampled trees. The log-likelihood contour is a cubic spline interpolation of the log likelihoods of all trees in the MDS plane; the MDS coordinates are computed from the BHV distances between trees. Each dot is a tree; the lighter the dot, the higher the likelihood of the tree. Right: the convex hull of the set of trees. The red dots are the vertices of the convex hull, displaying the sample trees on the boundary of the treespace.

provided in S1 Fig in S1 File. The Shepard diagrams and the correlations between tree distance and MDS coordinate distances for Figs 2 and 3 are given in Section 4.1 in S1 File.

## 2.5 Optimizing the branch-length of pathtrees

The pathtrees lay on the shortest path between a start and end tree (anchor trees). This path is constructed by refinement of the cone path as discussed in Section 2.1, so the resulting path-trees will more commonly have short branches. To find the best tree and also find potential local likelihood maxima, we need to find all distinct pathtree topologies and then optimize the

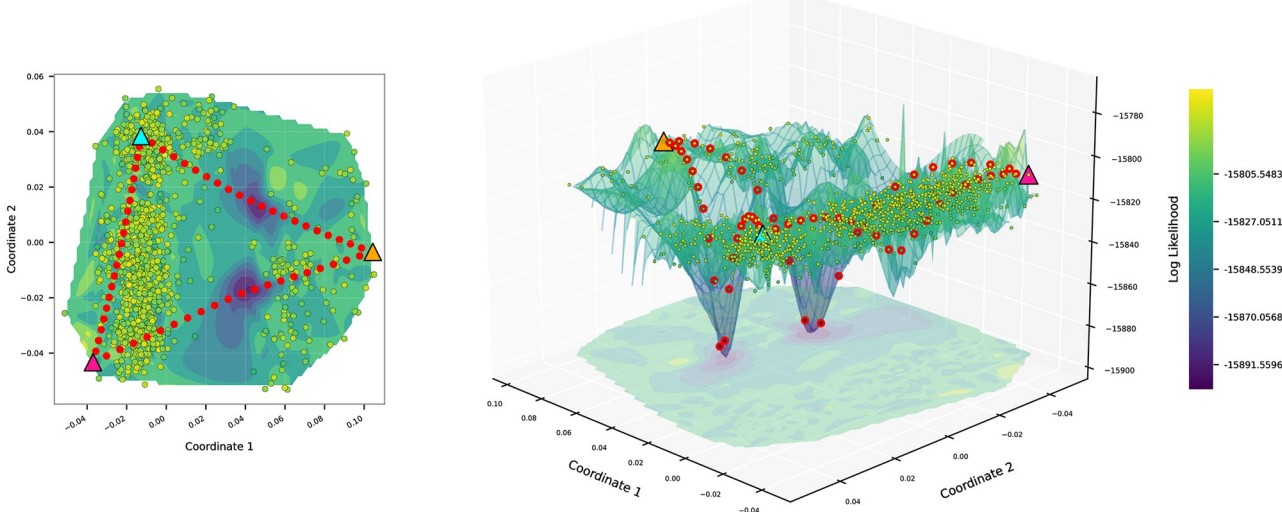

**Fig 3. An example of pathtrees between three arbitrary trees in the treespace.** Cubic spline interpolation of the log likelihood was used for the contour color (left) and the surface height (right) of the space inside the convex hull. 20 pathtrees (red dots) were generated on the shortest path between each pair of 3 trees (triangles).

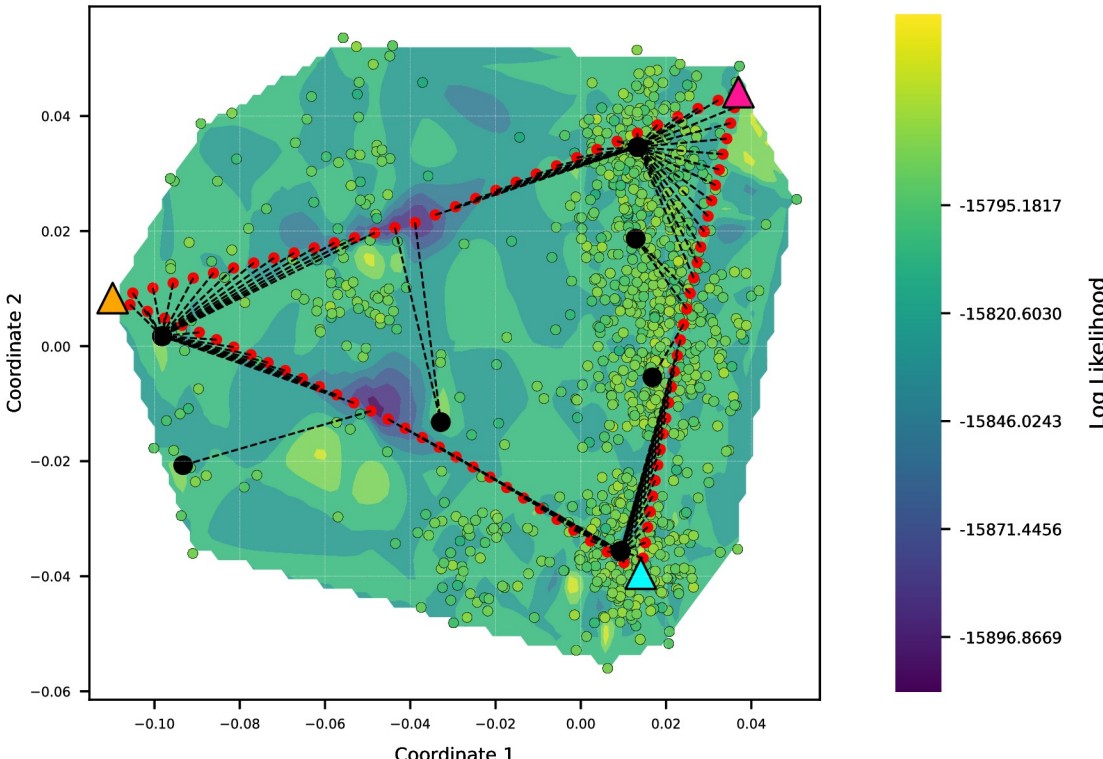

**Fig 4. An example of pathtrees and their corresponding optimized trees.** Each pathtree is connected to its corresponding optimized tree (black circles).

branch lengths for each different topology. We apply the unweighted Robinson-Foulds distance [23] to detect the different topologies; then we use PAUP* to optimize the branch lengths of a tree in each topology cluster. Fig 4 shows the locations of the pathtrees and their corresponding optimized trees. If the anchor trees have different topologies, they are located in different orthants. The pathtrees can have different intermediate topologies that, when optimized, will be located away from the shortest path. Unoptimized trees that start in low likelihood areas of treespace may move particularly far away, whereas trees that start on a ridge may not move far.

## 2.6 Software implementation

Our method is implemented in the Python package PATHTREES. The method uses the Java package GTP [20] to generate the geodesic between pairs of trees, the program PAUP* [11] for likelihood optimization, the Python modules DendroPy [53] for the Robinson-Foulds metric, and several other standard Python modules, such as SCIPY and NUMPY [54, 55].

We summarize the tree searching strategy of our package PATHTREES in the algorithm:

**Input**: Sequence data in PHYLIP format and $N$ rooted, non-ultrametric trees in plain Newick format sampled in connection with the sequence data (e.g. from a MCMC Bayesian analysis chain of the sequence data). The parameters $m$ and $n$ are the number of trees on the geodesic between each pair of starting trees and the number of trees to select with the highest likelihoods among the current sample trees, respectively.

**Output**: Trees on the shortest paths through treespace between all pairs of starting trees; branch-length optimized trees for different topologies; and MDS visualization of treespace and likelihood landscape inside the convex hall of input trees.

**Algorithm**:

1. Compute all pairwise distances between $N$ sample trees, and compute their MDS coordinates.

2. Extract the trees on the vertices of the convex hull of the $N$ sample trees in the 2-D MDS plane and consider them as starting trees.

3. Generate $m$ equally spaced trees on the geodesic between each pair of starting trees. Put all starting trees and generated pathtrees in the set current sample trees.

4. Calculate the likelihood of all current sample trees.

5. Select $n$ trees with the highest likelihood among current sample trees and classify them with respect to their topologies. Let there be $t$ topologies among them.

6. For each topology cluster, optimize the branch lengths for that topology and add these $t$ optimized trees to the current sample trees.

7. Visualize the contour and surface of the current sample trees by creating a distance matrix using either the weighted Robinson-Foulds or the BHV distance metric, interpolating log-likelihood values, and recomputing the MDS coordinates. The landscape and all current sample trees are colored according to the likelihood color bar. Then, the $n$ selected trees and the optimized trees are colored according to the purple spectrum color bar to distinguish different topologies. The lighter the purple color, the lower the likelihood value of the tree for that topology.

8. Return the current sample trees, the pathtrees, and the visualization; or continue with step 3 to zoom in the area of optimized trees of the current iteration.

## 2.7 Application to real data

We evaluate our approach PATHTREES using two datasets that were previously published: $D_1$ is a dataset of primates used in the tutorial for the program REVBAYES [56]. The dataset consists of 1141 base pairs of the mitochondrial cytochrome b gene of 23 primate taxa. $D_2$ is a larger mito-chondrial cytochrome b dataset of 182 milksnakes (1117 bp) [57, 58]. We chose the two differ-ent datasets because they represent very different situations. $D_1$ is a relatively small dataset but still too large to consider an exhaustive tree search. The species in the dataset are well defined, and the dataset contains enough variability to establish a phylogeny with branch lengths that are neither zero nor huge. In contrast, $D_2$ has about eight times more individuals than $D_1$; these individuals are only from a few species or subspecies, and many individuals share the same DNA sequence with others in the dataset. We deliberately did not remove individuals that have identical sequences, anticipating that the many zero branch lengths would be a stress test for our method.

For the primate dataset $D_1$, we collected 50,000 trees using the program REVBAYES. We used the instructions from the tutorial of REVBAYES [56], which are shown in Section 5 in S1 File. We selected every 38th tree from the last 3/4 of the total 50,000 sample trees (around 1000 sam-ple trees) and then extracted the trees on the vertices of the convex hull of these sample trees (14 trees) as starting trees for PATHTREES. These trees were the starting trees for two experi-ments: (1) 1 pathtree between each pair of starting trees and (2) a higher number of 15

pathtrees between each pair of starting trees to show a more detailed treespace. We used the BHV distance between trees for MDS for the first experiment and the faster wRF distance for the second experiment to make the computations tractable.

For the snake dataset $D_2$, we collected 10,000 trees using the program RevBayes. After removing the first 300 trees as burn-in, we selected every 20 trees and extracted 500 trees to be considered in convex hull analysis. The 14 trees on the vertices of the convex hull were used as starting trees in Pathtrees. For $D_2$, more complex dataset with 182 individuals, we deliberately gathered a smaller sample of 10,000 trees from RevBayes, in contrast to the primate dataset $D_1$. We burned in a minor portion of these trees to confirm that generating an extensive chain of RevBayes trees and a large burn-in is not necessary to achieve a satisfactory treespace for generating initial trees for Pathtrees. We computed 4 pathtrees along the shortest path between each starting tree pair and selected the 100 trees with the highest likelihood to classify by topology for branch length optimization. We then performed a second iteration of our method ("zoomed in") by computing the convex hull of the 100 optimized trees of the first iteration. The vertices of this convex hull became the starting trees for our second iteration, and in this iteration we computed 5 pathtrees along each shortest path between starting tree pairs. For both iterations, we used the wRF distance between trees as input for MDS, and used thin-plate spline interpolation to visualize the likelihood landscape.

## 2.8 Comparison of PATHTREES with heuristic tree searches

We compared the highest likelihood trees found by Pathtrees with those generated by the maximum likelihood software Paup* 4.0a (build 168) [11], RAxML 8.2.12 [9], and the Bayesian inference software RevBayes 1.1.1 [13]. These programs perform heuristic searches. Paup* and RAxML search will swap on new tree topologies until a local maximum has been reached and no new tree topologies need further evaluations. These heuristics do not guarantee to recover the global maximum likelihood tree but usually deliver good results [9]. In contrast, RevBayes uses Markov chain Monte Carlo to evaluate the posterior probability of a tree while collecting trees along a Markov chain. The run time is user-determined and needs to be long enough to sample good candidate trees. These trees are then used to estimate the maximum *a posteriori* tree.

We conducted several experiments to evaluate whether the number of intermediate pathtrees for each pair of anchor trees affects the accuracy of the MDS reconstruction of the likelihood surface and how well we can recover the best tree. We use the Jukes-Cantor mutation model for the likelihood calculation throughout all analyses. Using such a simple model reduces potential difficulties introduced by parameter fitting.

We compared Pathtrees with Paup* and RevBayes for the primate dataset $D_1$ because we were confident that the RevBayes analysis converged. For the snake dataset $D_2$, we compared Pathtrees with Paup* and RAxML because even long runs of RevBayes did not deliver stable results. Both Paup* and RAxML were run without improving parameters that tune the heuristic search. All generated pathtrees were compared with all the evaluated trees in RevBayes to investigate whether our approach can find topologies that were not visited by RevBayes.

## 3 Results

We use our methods for landscape generation, visualization, and finding high likelihood trees to examine the two datasets $D_1$ and $D_2$. Fig 5 shows the contour and surface plots of the visualized treespace landscape for the primate dataset $D_1$, generated with one pathtree per pair of starting trees (14 starting trees and 91 total pathtrees), and using the BHV distance for MDS input and thin-plate spline interpolation of the log-likelihood values of all plotted trees. Fig 5A

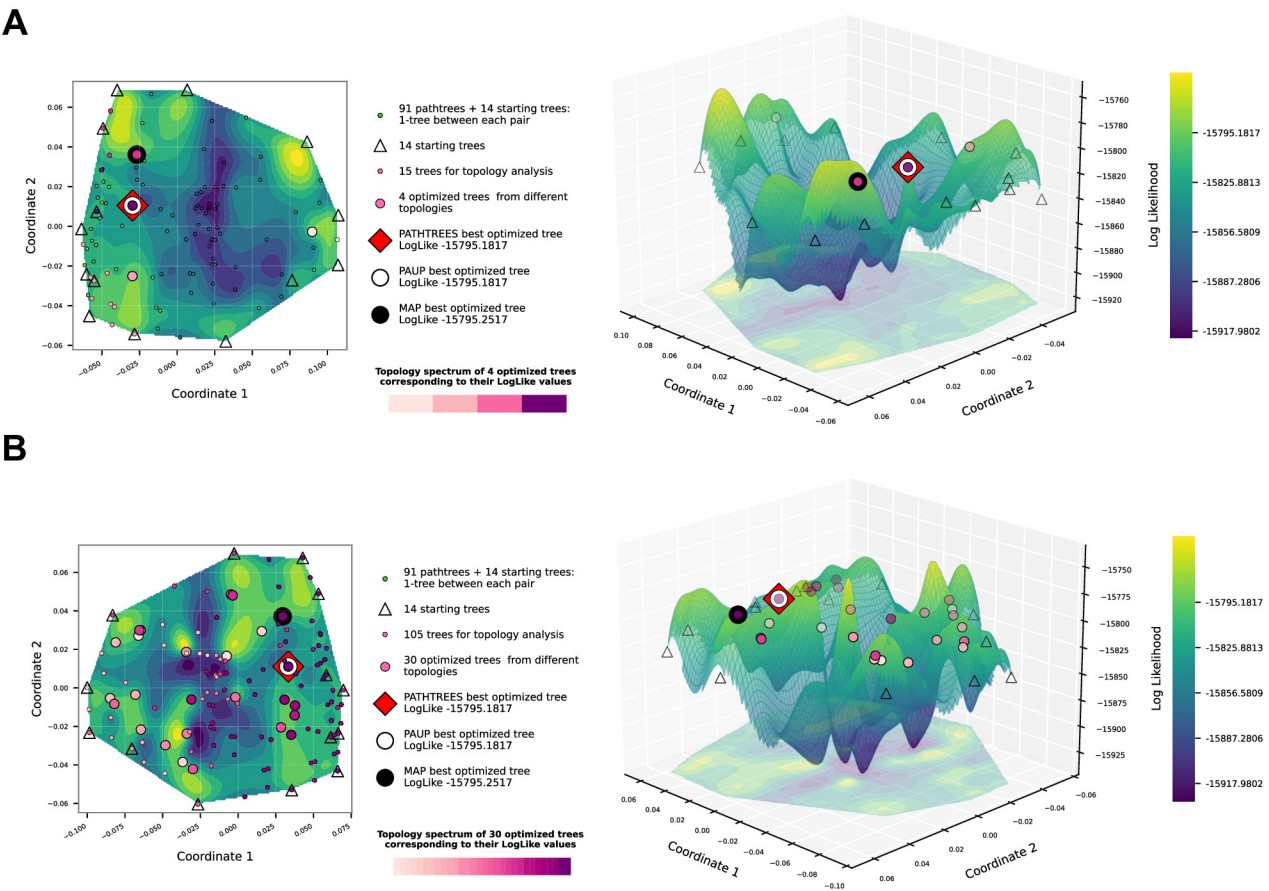

**Fig 5. Contour and surface plots of PATHTREES for dataset $D_1$ when generating one pathtree per starting tree pair (91 pathtrees), using BHV distances for the MDS input, and thin-plate spline interpolation for the landscape.** The vertices of the convex hull of the selected 1000 sample trees are the starting trees (14 triangles). (A) 15 trees with the highest likelihood were selected from the 91+ 14 trees and classified based on their topologies (4 topologies with the relative likelihoods of their branch-length optimized trees given by the purple spectrum). Each medium-size circle with a purple color shows one of these locally optimal trees. The red square shows the best likelihood tree in the treespace found by PATHTREES. The large white disk shows the best tree of PAUP*, which is identical to PATHTREES' optimal tree. One of PATHTREES' locally optimal trees matches the MAP tree (big black circle). (B) All 91+ 14 trees were selected and classified based on their topologies (30 topologies). Medium-size disks on a purple spectrum show the 30 locally optimal trees and their relative likelihoods. The big red square shows the best likelihood tree of PATHTREES.

shows the 15 trees with the highest likelihood (small pink/purple colored disks) selected from the 91 pathtrees +14 starting trees. These 15 trees contained 4 different topologies, and the branch lengths in each topology cluster were optimized (medium-size circles). These 15 trees are colored by their topology, according to the purple spectrum color bar. The lighter the purple color, the lower the likelihood value of the optimized tree for that topology. Among locally optimal trees associated with different topologies, we found the best tree with the log likelihood of −15795.1817 (red square). We find the same tree as PAUP* (big white circle) as the optimal tree and also find the same tree as the maximum Carlo to evaluate the posterior probability of a tree while collecting trees along a Markov chain. The run time is user-determined and needs to be long enough to sample good candidate trees. These trees are then used to estimate the maximum *a posteriori* tree (MAP) in REVBAYES (big black circle) as a locally optimal tree. In the plots, we see higher likelihood areas near the outside of the convex hull and a lower likelihood region in the middle. The 15 highest likelihood trees selected for topology analysis and their optimized topology trees are also towards the edges of the convex hull.

Fig 5B shows the same 91 pathtrees and 14 starting trees as Fig 5A, but optimized branch lengths have been computed for all their different topologies (30 topologies in total). Among all locally optimized trees (30 local optima corresponding to different topologies), we found again the best tree with the log likelihood of −15795.1817. In the plots, we still see a lower like-lihood region in the middle of the convex hull, along with some points of higher likelihood corresponding to optimized trees. The orientation of the landscape is flipped from Fig 5A, but this is a possibility with MDS.

Fig 6 shows the contour and surface plots of the visualized treespace landscape for dataset $D_1$, generated with 15 pathtrees per starting tree pair, and using the wRF distance for MDS input and thin-plate spline interpolation of the log-likelihood values of all plotted trees. We generated a total of 1365 pathtrees with 43 different topologies to show a detailed treespace with a high number of trees. After optimizing the branch lengths for all 43 topologies, PATH-TREES found the same best tree as PAUP*, as previously found, with the log likelihood of −15795.1817. With a higher number of pathtrees, we have a surface where trees are evenly spread out, and the gaps are filled. The best trees detected by PATHTREES, PAUP*, and REVBAYES were added to S2 Fig in S1 File. We still see a lower likelihood region in the middle of the plots. Pathtrees with the same topology are generally grouped together, and the sampling is dense enough that we can see the path of the geodesics (curved lines of pathtrees) in some cases.

For the snake dataset $D_2$, Fig 7 shows the contour and surface plots of the likelihood land-scape from two iterations of PATHTREES, with the second iteration zooming in on the top 100 optimized trees from the first iteration. Fig 7A shows the first iteration plots, generated using 4 pathtrees per shortest path between starting trees (14 starting trees) and selecting the 100 trees with the highest likelihood to classify by topology. All selected trees have different topologies, giving 100 optimized trees. PATHTREES found a similar tree to the best tree from PAUP*, both with the log likelihood −5225.5856. Fig 7B shows the second iteration plots, zooming in on the convex hull of the 100 optimized trees from the first iteration. The 11 trees on the vertices of this convex hull became the starting trees for the second iteration. After generating 5 pathtrees between each starting tree pair and classifying the generated pathtrees based on topology, we found 154 different topologies, which we optimized. Despite additional topologies, we found the same tree with highest likelihood as in the first iteration. In Fig 7A, we see only a small area of high likelihood trees, separated into two peaks, near the center. In Fig 7B, zoomed in on this high likelihood section, we still see a lower likelihood chasm dividing the convex hull. The optimized topology trees are near the edges of the convex hull.

A comparison of the three optimal trees, namely ours, the best PAUP* tree, and the best RAxML tree, revealed that all trees are different from each other. Table 1 shows the weighted and unweighted Robinson-Foulds distances between the trees and their log likelihoods. Our tree and the PAUP* tree are close and only differ by four splits, whereas the RAxML tree differs from both our tree and the PAUP* tree by nine splits. The wRF distances between these three trees show similar relationships. The RAxML tree is different because its topology was found by applying the option GTRCAT and constraining for the JC69 model (RAxML does not have an equivalent to the simple JC69 model without site rate variation). We then used PAUP* to find the optimal branch lengths for the RAxML topology, and the plain JC69 model to com-pute the likelihoods. The log likelihoods for all these 'best' trees are very similar, and compar-ing their location on the surface in Fig 7 also shows that the best PATHTREES and PAUP* trees are close, whereas the RAxML tree seems to be on a different local maximum on the surface. S3 Fig in S1 File contains all three trees showing their topology differences.

We used MDS to visualize the BHV treespace. This compression of the high dimensional space to 2 dimensions may lead to artifacts, or spurious errors, in the visualization. We vali-dated the accuracy of our visualizations using correlation analyses between the distance matrix

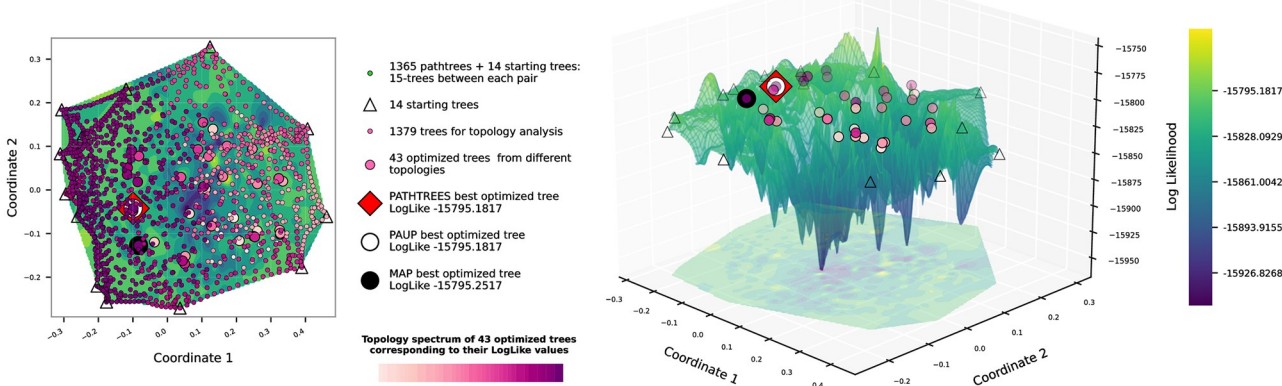

**Fig 6. Contour and surface plots of Pathtrees for dataset $D_1$ when generating 15 pathtree per starting tree pair, using weighted Robinson-Folds distances for the MDS input, and thin-plate spline interpolation for the landscape.** After optimizing branch lengths for the 43 different topologies (colored by relative likelihood of this optimized tree using the purple spectrum), Pathtrees found the best tree (red square) with the log likelihood −15795.1817, which is the same as the optimal tree of Paup* (white circle).

in BHV space and the MDS distance matrix using the first and second coordinates. For the primate dataset $D_1$, the Pearson correlation for Fig 5B was 0.9237 (additional correlation measures and the Shepard diagrams are shown in Section 4.1 in S1 File). For the snake dataset $D_2$, the Pearson correlation coefficient for Fig 7A (first iteration), was lower than for Fig 7B (second iteration), 0.6856 and 0.9702 respectively, because the first figure covers a much larger area of trees than the second (more correlation measures and the Shepard diagrams are shown in Section 4.2 in S1 File).

For the primate dataset $D_1$, there are 168 different topologies among 50,000 RevBayes trees, and for the snake dataset $D_2$, all 10,000 RevBayes trees are having different topologies. Our pathtrees are only based on the trees on the convex boundary, and one may wonder whether this reduces the chance to find relevant new trees. For dataset $D_1$, Fig 5B, Pathtrees starting out with 14 starting trees and generating 91 pathtrees found a total of 28 different topologies among 91 pathtrees, 4 of which were different from all topologies found by 50,000 sampled RevBayes trees. A comparison with all 10,000 sampled RevBayes trees for dataset $D_2$ revealed that our method found all 364 pathtrees with new topologies in Fig 7A (first iteration), all of which were different from topologies found by RevBayes. In Fig 7B (second iteration), Pathtrees found a total of 145 different topologies among 275 pathtrees, again all different from the total topologies found by 10,000 sampled RevBayes trees.

## 4 Discussion

Our approach uses the Billera-Holmes-Vogtmann treespace framework to generate and visualize treespace, including the likelihood landscape, in an area of interest, and to augment the search for the maximum likelihood tree by investigating global and local maxima found by our method in this area. While there are other programs and packages for MDS visualizations of treespace under various distance metrics, Pathtrees is the only recent tool to include comprehensive visualization of phylogenetic likelihood landscapes over an area of interest. Our method can be used at different scales (e.g., see Fig 7) to better understand the spatial relationship between the highest likelihood trees. For example, in Fig 7, the initial landscape shows the highest likelihood trees are close together within the MCMC searched area of treespace; zooming in allows us to see that these trees form two high likelihood ridges, with a lower-likelihood

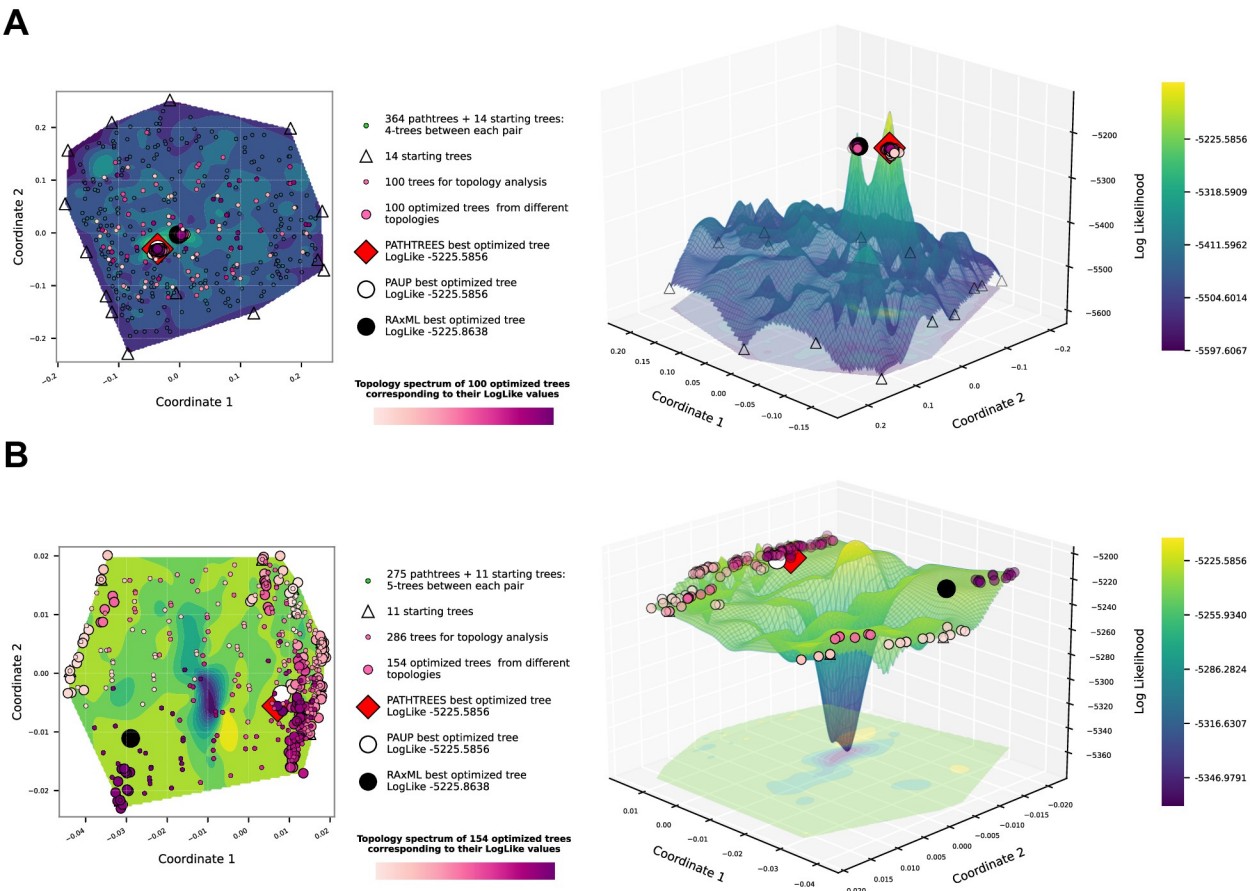

**Fig 7. Contour and surface plots from Pathtrees for dataset $D_2$, using weighted Robinson-Foulds distances for the MDS input, and thin-plate spline interpolation for the landscape.** (A) The vertices of the convex hull of the selected 500 sample trees are the starting trees (14 triangles). Four trees were generated on the geodesic of each pair of starting trees (364 pathtrees). Among them, 100 trees with the highest likelihood were selected to be classified based on topology. All 100 selected trees have different topologies (small circles, colored from the purple spectrum by the relative likelihood of their optimized tree). Each medium circle with a color from the purple spectrum shows the optimized tree with the corresponding topology. Among optimized trees, the red square shows the highest likelihood tree in the treespace, which is close but not identical to the best tree of Paup* (white circle). The RAxML tree (large black circle) is different. (B) This plot displays the treespace after zooming in on the optimized trees. The vertices of the convex hull of 100 optimized trees from the first iteration are the starting trees (11 triangles). Five pathtrees were generated between each starting tree pair (275 trees) and then all of them were selected to be classified by topology.

**Table 1. Comparison of the best tree found by Pathtrees, Paup*, and RAxML using the data $D_2$.**

| Tree | uRF / wRF | | | ln L |
|------|-----------|------|------|------|
| | **Pathtrees** | **Paup*** | **RAxML** | |
| Pathtrees | - | 4 | 9 | -5225.5856 |
| Paup* | 0.0054 | - | 9 | -5225.5856 |
| RAxML | 0.0343 | 0.0325 | - | -5225.8638 |

Above the diagonal is unweighted Robinson-Foulds (uRF) for all pairs and below the diagonal is the weighted Robinson-Foulds distance (wRF). The last column is the log likelihood for each tree.

region between them. Secondly, our method can find trees with the same maximal likelihood as other tree search programs, including one with a different topology for dataset $D_2$. These results suggest PATHTREES could potentially be used to understand treespace islands and terraces better.

A limitation of our method is the need for starting trees, due to the enormity of treespace. For our two example datasets, treespace has $5.6 \times 10^{26}$ orthants and $4.7 \times 10^{384}$ orthants, respectively, so we cannot start with the full treespace and just zoom in. We decided to use a sample of trees from a Bayesian phylogenetic inference program, REVBAYES. In principle, any set of reasonably close trees to the best tree may work as a starting point. Using the convex hull in MDS space to start our approach helped reduce the geodesic distance matrix size used to create the visualizations. In a way, we treat the convex hull in MDS space as an approximation of the convex hull in treespace [59, 60]. However, we believe that the hull defined by MDS allows us to investigate good and best trees within its boundary. In our examples, the best tree, found by other procedures and ours, is within this MDS hull.

A second limitation of our method is that tree topologies along geodesics can only contain splits that are in one of the two endpoints trees and contain all splits that are in both of the endpoint trees. However, we argue this constraint is not onerous—there can still be exponentially many topologies fitting this description—but means that the intermediate tree topologies explored are relevant. The successful coalescent-based species tree estimation program ASTRAL had a similar split constraints [61]. Additionally, trees on a geodesic between two starting trees are unlikely to be in the set of best trees because this geodesic defines their branch lengths and topology. Thus, our method takes these pathtrees as starting points and optimizes their branch lengths (once per topology). This procedure allows us to describe the tree landscape along the path and describe the maxima for specific topologies. Using many pathtrees that were optimized or not optimized will give a precise picture of the tree landscape (for example, Figs 5–7). Our method provides a simple framework for exploring landscapes of phylogenetic trees and visualizing their relationship in a continuous and low-dimensional projection facilitated by multidimensional scaling and an interpolation method, cubic spline or thin-plate spline interpolation, to reveal potential tree islands. The visualization gives a good impression of the likelihood landscape: general patterns can be shown with few trees, but details may need many more trees to create a more smooth surface. However, even with many trees, the visualization may contain artifacts in areas where there are no trees, for example, the spikes in Fig 7.

A Bayesian inference method evaluates trees according to the posterior probability, which is dominated by the likelihood of the tree when we assume vague prior distributions. It is fair to say that even a long inference run will not explore all possible topologies. Even our small dataset of 23 primate species has too many different topologies to explore all in a Bayesian context in a reasonable time. Of course, most of these topologies will have an inferior fit to the sequence data, but even those trees that fit the data relatively well are many. In contrast to Bayesian inference and heuristic search methods, our method does not depend on an optimality criterion to pick trees that lie on the shortest path between two arbitrarily chosen trees. This allows exploring topologies that were never visited with a good Bayesian run or any other heuristic search as we have shown.

We picked the two datasets because they represent very different situations. The primate dataset $D_1$ is relatively small. However, it is still too big to be solved exhaustively. It provides many mutational differences allowing good resolution of branch lengths and branching patterns. The snake dataset $D_2$ has eight times more individuals that are closely related. Many sequences are identical, leading to many multifurcations.

Heuristic searches for $D_1$ and $D_2$ are fast, and even an MCMC run with REVBAYES does not need a long time for $D_1$. However, we had difficulties estimating a MAP tree for $D_2$ because we

had difficulty running to convergence. Pathtrees generates independent trees, evenly spaced along geodesics, to help visualize treespace, find optimal trees, and explore the likelihood surface near these optimal trees. Pathtrees optimizes its pathtrees and finds local maxima for the evaluated topologies; these are the same as those found by Paup* and RevBayes for $D_1$. Interestingly, the MAP tree and the Paup* tree differ by two splits but when the branch lengths are optimized deliver log likelihoods that are very similar. Even ten times longer runs in RevBayes deliver the same MAP tree. So while it may seem difficult for RevBayes to explore that particular topology with the highest likelihood, the difference is only 0.07 log units. We certainly would not exclude the MAP tree in a likelihood ratio test. The snake dataset $D_2$ reveals that many trees will be good candidates for the best likelihood tree. The Paup*, RAxML, and Pathtrees best trees have all different topologies but very similar log likelihoods. These trees are also very similar, with only 4 or 9 different splits between them. Pathtrees helps give insights about the likelihood surface, such that it is rather flat and therefore will have many potential trees with similar likelihoods.

We believe that our method, implemented in Pathtrees, complements heuristic search phylogenetic analyses and allows visualization of the treespace and finding alternative trees with log likelihoods that are potentially better than those of heuristic searchers. For example, a new way to propose topologies for tree search could be by sampling pathtrees along a geodesic between two trees, or in a region, of interest. Alternatively, pathtrees could become starting trees themselves for a maximum likelihood search. The visualization of the likelihood surface also allows the discussion of local likelihood maxima, which we hope will lay the groundwork for improving search algorithms.

## Supporting information

**S1 File. Supporting information for: Geodesics to characterize the phylogenetic landscape.**
(PDF)

## Acknowledgments

We thank two anonymous reviewers for their comments and suggestions.

## Software availability

We implemented our method as free software named Pathtrees under the MIT open-source license. The source code and the documentation of Pathtrees are available at https://github.com/TaraKhodaei/PATHTREES.git.

## Author Contributions

**Conceptualization:** Marzieh Khodaei, Megan Owen, Peter Beerli.

**Data curation:** Marzieh Khodaei, Peter Beerli.

**Formal analysis:** Marzieh Khodaei, Megan Owen, Peter Beerli.

**Funding acquisition:** Megan Owen, Peter Beerli.

**Investigation:** Marzieh Khodaei, Megan Owen, Peter Beerli.

**Methodology:** Marzieh Khodaei, Megan Owen, Peter Beerli.

**Project administration:** Marzieh Khodaei, Peter Beerli.

**Resources:** Marzieh Khodaei, Megan Owen, Peter Beerli.

**Software:** Marzieh Khodaei, Megan Owen, Peter Beerli.

**Supervision:** Marzieh Khodaei, Megan Owen, Peter Beerli.

**Validation:** Marzieh Khodaei, Megan Owen, Peter Beerli.

**Visualization:** Marzieh Khodaei, Peter Beerli.

**Writing – original draft:** Marzieh Khodaei, Megan Owen, Peter Beerli.

**Writing – review & editing:** Marzieh Khodaei, Megan Owen, Peter Beerli.

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
