## [Decision Letter · Decision Letter 0]

27 Apr 2023

PONE-D-23-08402Geodesics to Characterize the Phylogenetic LandscapePLOS ONE

Dear Dr. Khodaei,

Thank you for submitting your manuscript to PLOS ONE. After careful consideration, we feel that it has merit but does not fully meet PLOS ONE’s publication criteria as it currently stands. Therefore, we invite you to submit a revised version of the manuscript that addresses the points raised during the review process. Both referees recommended minor edits.  Thus, please submit respond to each of their comment with the edited manuscript. 

We look forward to receiving your revised manuscript.

Kind regards,

Ruriko Yoshida

Academic Editor

PLOS ONE

Journal Requirements:

3. Please update your submission to use the PLOS LaTeX template. The template and more information on our requirements for LaTeX submissions can be found at http://journals.plos.org/plosone/s/latex

Additional Editor Comments (if provided):

Both referees recommended minor edits. So please respond their comments and submit with the edited manuscript.

Reviewers' comments:

Reviewer's Responses to Questions

**Comments to the Author**

1. Is the manuscript technically sound, and do the data support the conclusions?

Reviewer #1: Yes

Reviewer #2: Yes

2. Has the statistical analysis been performed appropriately and rigorously? 

Reviewer #1: Yes

Reviewer #2: Yes

3. Have the authors made all data underlying the findings in their manuscript fully available?

Reviewer #1: Yes

Reviewer #2: Yes

4. Is the manuscript presented in an intelligible fashion and written in standard English?

Reviewer #1: Yes

Reviewer #2: Yes

5. Review Comments to the Author

Reviewer #1: This is a very nicely written paper that was a pleasure to read. As far as I can tell the results are interesting, and novel, and likely to be of interest to those in the field of computational phylogenetic inference. The comments below are all of a minor nature:

line 19-20. Finding trees not found with MCMC could be spurious - it's not obviously a good thing, though the authors make a strong case in the Discussion. maybe a few more words in the abstract, or even just use "relevant trees".

line 51-52. The word "gappy" is obviously not technical and perhaps means something clear to the authors. It wasn't quite clear to me. Is this all with respect to BHV space? Likewise "some regions", refers to BHV space?

line 89-90. This reads as though the sampling is across *all* geodesics, which is clearly not intended. Also "area of interest" is used a few times around here (line 89, 92, 115, 119, 121, 155, 173), and it wasn't clear at this point where this came from. Much later it is explained, but here it is opaque. Maybe somewhere explain that attention is focussed on an area of interest to explore, to motivate all this reference to it.

line 120. The "potentially" is doing a lot of work here. Sounds uncertain. Is there reason to hope?

line 193. Typo: "distance" matrix

line 206. There is discussion about the closeness of trees in a 2D compression of tree-space. But how does this relate to actual tree-space? Trees close in 2D may be far apart in BHV space. But are trees far apart in 2D necessarily far apart in BHV? Does this compression have any nice isometric properties that mean it tells us something about the real space?

line 241. Where does the m come from? It's not an algorithm input.

line 244. Likewise where does n come from?

line 267. Why 3/4? Earlier it was 1/10.

line 273, 277. I don't understand why a smaller sample of 10,000 trees was taken when the data set has more taxa than D1. Also, for consistency, in line 277 use a comma in "10000".

line 296. Italics for a posteriori?

line 424. "lie" instead of "lay"?

Reviewer #2: Main Ideas:

This work aims to provide a new visualization method for phylogenetic trees that allows for the construction of intermediate trees connecting two given trees. They do this by mapping treespace into lower dimensions means representing a high-dimensional space (space of all possible trees) in a lower-dimensional space (e.g., a 2D or 3D space) to make it easier to visualize and explore. The authors provide a new method for constructing intermediate trees between two trees in BHV space.

The proposed visualization method is effective and potentially has applications exceeding the scope of this work.

This work focuses on applications to ML and Bayesian tree searches. The authors compare it with existing methods such as PAUP*, RAxML, and RevBayes. They use primate and milk snake datasets and show that their method can find trees that were not found with heuristic search methods.

General Feedback:

The paper is well-written and informative from both the mathematical and biological perspectives. Given the readily available software, the results are new and likely to be used. It is a notable f interdisciplinary project combining data science, mathematics, and biology.

However, the most significant limitation of the current version of the article is that the audience needs to be made more explicit. Without framing the implications of this work, it is likely to be encountered by mathematicians who wish to know more about the geometry of the mapping from BHV space to R2, And by practitioners who may want to know the exact scenarios under which they should be using this tool and to have more evidence that it can be relied upon to be helpful.

However, doing either of these, much less both, would be more work than is appropriate for a revision and would likely warrant separate publications. So instead, I recommend this paper be accepted with relatively minor revision per the suggestions below.

Concerns:

• Revise the framing of the introduction in a manner that clearly sets out what a reader can expect from the paper in terms of usability and mathematical nuance.

• There should be more discussion regarding potential use for biologists both now and in the future. The software is there, but it needs to be advertised more in the paper so people will use it.

Line 357-363,

• This statement is troubling because one of the more significant results is that the pathtree method found an optimal tree that another method missed. However, it needs to be clarified if RAxML missed this more optimal tree intrinsically or missed it due to the switching between models.

Minor Concerns:

• The authors could provide a picture of the "gappy" space to help readers understand the problem. The term might also confuse someone who thinks it refers to the sequences having many gaps.

• The authors should clarify technical terms (such as topology proportions, consistency of generated treespace, and artifact). It sometimes took work to tell when these were being used formally or informally.

• The authors should explain more about the clustering results and compare them with RF distance results.

• The authors could provide more details on the relationship between the BHV space and the MDS plane. In particular, there are choices of metrics in multiple stages of the Algorithm, and it needs to be clarified what the implications of the choices may be.

• Line 197: Get rid of "using an interpolation method" since that is clarified in the following sentence.

• Line 185 is "not" an algorithm. Also, move this sentence above the example using REV Bayes.

Using distance methods as an example of searching through tree space as an optimization problem seems like an alternative theoretical problem to that described here.

• Namely, in the ML and Bayesian Contexts, you search through trees and check a function's value. In contrast, in distance-based methods, one uses finite data to build a tree, typically in some sequential order, to (potentially) optimize something, thus avoiding even the intention of a full global search. Readers will likely understand this work more clearly in the context of the ML/Bayesian methods.

• Section 1.5 Explain why we need path trees with the same topology. If I understand this section correctly, it looks like the optimized trees (same topology as the two consecutive path trees) can appear quite far away in this diagram.

• Line 121: Points "on" a convex Hull. Does that mean points on the boundary of the convex hull, vertices, or any points in the convex hull?

• Line 263 Consolidate the dataset into unique site patterns. Does this mean not only using SNPS or not excluding individuals whose sequences match?

Line 307.

• Using "our method twice" in this sentence is not needed.

• In the results section, it might be easier to refer to D_1 and D_2 as the primate and milk snake datasets.

• Line 346: Should it be "The 11 trees"

• In terms of your figures, you indicate that the path trees correspond in clusters to tree topologies. It would be interesting to see how tree topology differences map into the R2 space as an overlay in these examples or toy mapping. The reason is that there would be more interest biologically, in very close local maxima that correspond to differing topologies, than in local maxima corresponding to different branch lengths.

• In the correlation coefficient section, Page 16, it would be helpful to have a reference indicating that the Person coefficients or Shepard diagrams are standard measurements when reducing the data dimension.

• Paragraphs 373-380 would help interpret the results to know how many different topologies were among the REVBayes trees.

• Line 383: I love the term "augment an ML tree search" Perhaps this term is the key to clearly framing the paper.

• Paragraph 403, I am not sure about the latest version of Astral, but earlier versions had the limitation of requiring the splits of the species tree to be among the set of splits of the gene trees. They showed that this is not an issue in practice or even theoretically. They do not need to redo their result; putting this in a familiar context for a user is helpful.

• In this mapping, it may be helpful to pose some of the open questions regarding the relationship between BHV space and R^2.

6. PLOS authors have the option to publish the peer review history of their article (what does this mean?). If published, this will include your full peer review and any attached files.

Reviewer #1: No

Reviewer #2: No

---

## [Author Response · Author response to Decision Letter 0]

24 May 2023

Dear Ruriko Yoshida,

Thank you for the opportunity to revise and resubmit our manuscript “Geodesics to characterize the phylo- genetic landscape”. We appreciate you and the reviewers for your careful review and valuable comments. The authors have carefully considered the comments. We believe that the reviewer’s insight has substantially improved our manuscript. We hope the revised manuscript meet your high standards.

We provided the following items when submitting our revised manuscript:

• Response to Reviewers: All our answers are in blue color.

• Revised Manuscript with Track Changes: A marked-up copy of the Manuscript with track changes. All added sections are in green color, and all deleted sections are crossed out in red color.

• Revised Supporting Information with Track Changes: A marked-up copy of the Supporting Infor- mation with track changes. All added sections are in green color, and all deleted sections are crossed out in red color.

• Manuscript: An unmarked version of the Manuscript without tracked changes. We updated the Manuscript to the PLOS LaTeX template. We also submit the LaTeX format of our Manuscript.

• Supporting Information: An unmarked version of the Supporting Information without tracked changes.

• Figures: Our package PATHTREES generates figures in the pdf format. Based on PLOS ONE figure requirement, we provided tiff format of figures using the PACE conversion tool in PLOS. We prefer to use pdf format of figures since they have good resolution that users can zoom in.

All authors have read and approved the final version of the manuscript. 

Sincerely,

Marzieh Khodaei

---

## [Decision Letter · Decision Letter 1]

5 Jun 2023

Geodesics to Characterize the Phylogenetic Landscape

PONE-D-23-08402R1

Dear Dr. Khodaei,

We’re pleased to inform you that your manuscript has been judged scientifically suitable for publication and will be formally accepted for publication once it meets all outstanding technical requirements.

Kind regards,

Ruriko Yoshida

Academic Editor

PLOS ONE

Additional Editor Comments (optional):

Both referees are happy with the revised manuscript. Thus, I recommend to publish this paper in the journal.

Reviewers' comments:

Reviewer's Responses to Questions

**Comments to the Author**

1. If the authors have adequately addressed your comments raised in a previous round of review and you feel that this manuscript is now acceptable for publication, you may indicate that here to bypass the “Comments to the Author” section, enter your conflict of interest statement in the “Confidential to Editor” section, and submit your "Accept" recommendation.

Reviewer #1: All comments have been addressed

Reviewer #2: All comments have been addressed

2. Is the manuscript technically sound, and do the data support the conclusions?

Reviewer #1: Yes

Reviewer #2: Yes

3. Has the statistical analysis been performed appropriately and rigorously? 

Reviewer #1: Yes

Reviewer #2: Yes

4. Have the authors made all data underlying the findings in their manuscript fully available?

Reviewer #1: Yes

Reviewer #2: Yes

5. Is the manuscript presented in an intelligible fashion and written in standard English?

Reviewer #1: Yes

Reviewer #2: Yes

6. Review Comments to the Author

Reviewer #1: The authors have addressed all of my comments well. The only further thing I found to address is a missing fullstop on line 451 in the Discussion.

Reviewer #2: The authors took time and care to address all of the concerns. In each instance they either made an appropriate change or provided a reasonable explanation. They authors should be commended for their careful work.

7. PLOS authors have the option to publish the peer review history of their article (what does this mean?). If published, this will include your full peer review and any attached files.

Reviewer #1: No

Reviewer #2: No

---

## [Editor Report · Acceptance letter]

13 Jun 2023

PONE-D-23-08402R1 

Geodesics to Characterize the Phylogenetic Landscape 

Dear Dr. Khodaei:

I'm pleased to inform you that your manuscript has been deemed suitable for publication in PLOS ONE. Congratulations! Your manuscript is now with our production department. 

Kind regards, 

on behalf of

Dr. Ruriko Yoshida 

Academic Editor

PLOS ONE